# Magnitude, distribution and determinants of non-utilization of antenatal care services among women in low- and middle-income countries: Insights for implementation of WHO recommendations

Tadesse Tarik Tamir[ID][1]*, Deresse Abebe Gebrehana[2], Alebachew Ferede Zegeye[3], Bewuketu Terefe[4], Berhan Tekeba[1]

1 Department of Pediatrics and Child Health Nursing, School of Nursing, College of Medicine and Health Sciences, University of Gondar, Gondar, Ethiopia, 2 Department of Internal Medicine, School of Medicine, College of Medicine and Health Sciences, University of Gondar, Gondar, Ethiopia, 3 Department of Medical Nursing, School of Nursing, College of Medicine and Health Sciences, University of Gondar, Gondar, Ethiopia, 4 Department of community Health Nursing, School of Nursing, College of Medicine and Health Sciences, University of Gondar, Gondar, Ethiopia

* tadestar140@gmail.com

## Abstract

### Introduction

Pregnancy is a pivotal stage that fosters health and prepares women and their families for the transition to parenthood. Antenatal care (ANC) encompasses the services provided by skilled healthcare professionals to pregnant women and adolescent girls, ensuring optimal health outcomes for both mother and child throughout pregnancy. The majority of maternal and child deaths occur in low- and middle-income countries (LMICs), where access to essential healthcare services, including ANC, remains a significant challenge. This study was aimed to assess the magnitude, spatial distribution, and determinants of non-utilization of ANC services in LMICs.

### Methods

This study analyzed data from the most recent Demographic and Health Surveys conducted between 2015 and 2024, encompassing a total of 47 LMICs. The analysis included a weighted sample of 480,068 women. We employed spatial analysis to illustrate the geographic distribution of non-utilization of antenatal care and hierarchical analysis to identify contributing factors. ArcGIS 10.8 and Stata 17 were utilized for spatial and hierarchical analysis, respectively. Adjusted odds ratios with 95% confidence intervals (CIs) were calculated, and factors were considered statistically significant at a p-value of less than 0.05.

**Data availability statement:** All relevant data are within the paper and its Supporting Information files.

**Funding:** The author(s) received no specific funding for this work.

**Competing interests:** No authors have competing interests.

## Results

Pooled magnitude of non-utilization of ANC among women in LMICs was at 10.59%, ranging from 40.05% in Afghanistan to 0.76% in Burundi, with many regions in several countries identified as hotspots for ANC non-utilization. Factors significantly associated with higher odds of non-utilization included having no (AOR = 3.28; 95% CI: 3.02–3.55) or low (primary schooling: AOR = 1.81; 95% CI: 1.67–1.96, and secondary schooling: AOR = 1.28; 95% CI: 1.18–1.38) education, being unmarried (AOR = 1.35; 95% CI: 1.29–1.41), lower wealth index (poorest: AOR = 1.87; 95% CI: 1.77–1.98), poorer: AOR = 1.45; 95% CI: 1.38–1.54, middle: AOR = 1.17; 95% CI: 1.11–1.24, and richer: AOR = 1.09; 95% CI: 1.04–1.15), having no media exposure (AOR = 1.68; 95% CI: 1.64–1.73), residing in rural areas (AOR = 1.05; 95% CI: 1.02–1.09), facing distance issues to health facilities (AOR = 1.31; 95% CI: 1.28–1.34), and the low-income level of the countries (AOR = 2.27; 95% CI: 1.23–6.74).

## Conclusion

A significant proportion of women in LMICs have not utilized antenatal care services. Factors at the individual, community, and country levels contribute to this non-utilization. Policymakers should focus on addressing these barriers to achieve the WHO recommendation of eight or more ANC contacts in LMICs.

## Introduction

Pregnancy is a crucial period for promoting good health and preparing women and their families both psychologically and emotionally for parenthood [1]. ANC refers to the care provided by skilled healthcare professionals to pregnant women and adolescent girls to ensure optimal health conditions for both mother and baby during pregnancy [2]. The World Health Organization (WHO) envisions a world where every pregnant woman and newborn receives quality care throughout pregnancy, childbirth, and the postnatal period. ANC is a crucial component of reproductive health care, offering a platform for health promotion, screening, diagnosis, and disease prevention [2,3]. Evidence-based practices implemented in a timely and appropriate manner through ANC can save lives [4]. Moreover, ANC provides an opportunity to communicate with and support women, families, and communities during a critical period in a woman's life [2,4]. High-quality antenatal care services enhance the survival and health of both mothers and babies [2,3]. Additionally, ANC offers women the opportunity to communicate with their healthcare providers, thereby increasing the likelihood of utilizing a skilled birth attendant [2,3].

In 2016, as the Sustainable Development Goals (SDGs) era began, preventable pregnancy-related morbidity and mortality remained unacceptably high [5]. In 2015, around 303,000 women and adolescent girls lost their lives due to complications related to pregnancy and childbirth [6]. That same year, 2.6 million babies were stillborn [6]. The vast majority of these maternal deaths (99%) and child deaths (98%)

took place in low- and middle-income countries [6]. Many of these maternal deaths could have been avoided if the women had access to quality ANC [3,7].

Previous studies have indicated that the use of antenatal care is affected by various factors. These include individual factors (such as socio-economic and reproductive characteristics), household or interpersonal factors (such as women's autonomy, husband's attitude and support, and family income), and health service factors (such as distance, accessibility, and availability) [8–11].

Despite significant advancements in expanding access to health services in low- and middle-income countries, the quality and access of care provided varies greatly across different countries and health conditions [12,13]. This inconsistency hampers progress in enhancing health outcomes [13]. This study aims to assess the magnitude, geographic distribution, and contributing factors of non-utilization of ANC services in LMICs, using data from the most recent demographic and health surveys conducted between 2015 and 2024. The findings will provide valuable insights for policymakers by identifying the extent of the problem, detecting high-risk areas, and pinpointing contributing factors to better address the needs of underserved populations.

## Methods

### Study area, data source and study period

This study focuses on LMICs and analyzed data from the most recent Demographic and Health Surveys (DHSs) conducted between 2015 and 2024. These datasets were accessed through the Monitoring and Evaluation to Assess and Use Results Demographic and Health Survey (MEASURE DHS) program. A total of 47 LMICs have nationally representative DHS data available within this timeframe (see Table 1 for a complete list of countries). However, spatial distribution analyses were conducted for only 41 of these countries due to the availability of GPS data, while the magnitude and determinants of non-utilization of antenatal care services were assessed for all 47 countries. We specifically utilized Individual Recode (IR) data and GPS data extracted from the DHS datasets. The IR data provided detailed individual-level health and demographic information, while the GPS data allowed for geographic analysis of health disparities.

### Population, sampling procedure and sample size

The source population for this study consisted of women aged 13–49 years in 47 LMICs who gave birth five years preceding the survey. The study population, on the other hand, comprised women aged 13–49 years who had given birth within the three years preceding the survey and were residing in the enumeration areas covered by the surveys. The DHSs employed a stratified two-stage cluster sampling design. In the first stage, a probability proportional to size sample of enumeration areas is chosen. In the second stage, within the selected enumeration areas, a systematic sample of households is drawn [14]. We employed the svyset command to consider the complex survey design, making adjustments for individual sampling weights, clustering, and stratification present in the data [14]. Accordingly, a total weighted sample of 480,068 women were included in the analysis of this study (Table 1). Spatial analysis was conducted using ArcGIS 10.8, while hierarchical analysis was performed with Stata 17.

### Variables of the study

The outcome variable for this study is the non-utilization of ANC services. This variable was determined based on the ANC contact status of the women surveyed. Women were categorized as having non-utilization of ANC services if they had no recorded ANC visits throughout their pregnancy. Conversely, women who had at least one ANC contact were classified as utilizers of ANC services.

The explanatory variables selected for this study were based on established guidelines and relevant scholarly literature [15–17]. These variables were categorized into three levels. At the individual level, factors included age, education

**Table 1. Study setting, DHS year and sample size of non-utilization of ANC services in LMICs, 2015-2024 (n = 480,068).**

| Country | Survey year | Weighted Frequency (n) | Percent (%) |
|---|---|---|---|
| Afghanistan (NGPS) | 2015 | 19,609 | 4.08 |
| Albania | 2017−18 | 2,191 | 0.46 |
| Angola | 2015−16 | 8,495 | 1.77 |
| Armenia | 2015−16 | 1,361 | 0.28 |
| Bangladesh | 2022 | 5,104 | 1.06 |
| Benin | 2017−18 | 9,031 | 1.88 |
| Burkina Faso | 2021 | 6,400 | 1.33 |
| Burundi | 2016−17 | 8,941 | 1.86 |
| Cambodia | 2021 | 4,349 | 0.91 |
| Cameroon | 2018 | 6,613 | 1.38 |
| Colombia (NGPS) | 2015 | 9,370 | 1.95 |
| Cote d'Ivoire | 2021 | 5,301 | 1.10 |
| Ethiopia | 2016 | 7,590 | 1.58 |
| Gabon | 2019−21 | 4,370 | 0.91 |
| Gambia | 2019−20 | 5,372 | 1.12 |
| Ghana | 2022 | 4,706 | 0.98 |
| Guinea | 2018 | 5,488 | 1.14 |
| Haiti | 2016−17 | 4,890 | 1.02 |
| India | 2019−21 | 174,947 | 36.44 |
| Indonesia (NGPS) | 2017 | 15,016 | 3.13 |
| Jordan | 2023 | 3,657 | 0.76 |
| Kenya | 2022 | 9,474 | 1.97 |
| Lesotho | 2023−24 | 1,408 | 0.29 |
| Liberia | 2019−20 | 4,026 | 0.84 |
| Madagascar | 2021 | 9,232 | 1.92 |
| Malawi | 2015−16 | 13,515 | 2.82 |
| Maldives (NGPS) | 2016−17 | 2,368 | 0.49 |
| Mali | 2018 | 6,623 | 1.38 |
| Mauritania | 2019−21 | 7,705 | 1.61 |
| Mozambique | 2022−23 | 5,450 | 1.14 |
| Myanmar | 2015−16 | 3,581 | 0.75 |
| Nepal | 2022 | 2,765 | 0.58 |
| Nigeria | 2018 | 21,911 | 4.56 |
| Pakistan | 2017−18 | 6,711 | 1.40 |
| Papua New Guinea (NGPS) | 2016−18 | 6,636 | 1.38 |
| Philippines | 2022 | 4,146 | 0.86 |
| Rwanda | 2019−20 | 6,302 | 1.31 |
| Senegal | 2023 | 5,261 | 1.10 |
| Sierra Leone | 2019 | 7,326 | 1.53 |
| South Africa | 2016 | 3,036 | 0.63 |
| Tajikistan | 2017 | 4,395 | 0.92 |
| Tanzania | 2022 | 5,902 | 1.23 |
| Timor-Leste | 2016 | 5,000 | 1.04 |
| Turkey (NGPS) | 2019 | 2,032 | 0.42 |
| Uganda | 2016 | 10,152 | 2.12 |
| Zambia | 2018 | 7,325 | 1.53 |

*(Continued)*

**Table 1.** (Continued)

| Country | Survey year | Weighted Frequency (n) | Percent (%) |
|---------|-------------|------------------------|-------------|
| Zimbabwe | 2015 | 4,988 | 1.04 |
| **Total** | **2015-2024** | **480,068** | **100.00** |

NGPS: No GPS data

level, marital status, employment status, parity, number of living children, sex of the household head, household size, and wealth index. At the community level, we examined variables such as type of residence, distance to health facilities, and community development index. At the country level, we included income level, literacy rate, and geographical region. Additionally, we used DHS year as a control variable to handle the possible differential time effect from using DHS data of different time periods (Fig 1).

## Operational definition

**Wealth index**: In the DHS, households are categorized based on the quantity and types of consumer goods they own, which may include items such as televisions, bicycles, and cars, as well as housing features like sources of drinking water, bathroom facilities, and flooring materials. To compute these scores, principal component analysis is utilized. National wealth quintiles are established by assigning the household score to each usual household member, ranking individuals according to their scores, and dividing the population into five equal groups (poorest, poorer, middle, richer, and richest), each representing 20% of the total population [14,18].

   **Community development index:** The Community Development Index is informed by the Human Development Index (HDI), which emphasizes the importance of access to essential services such as electricity, clean drinking water, and

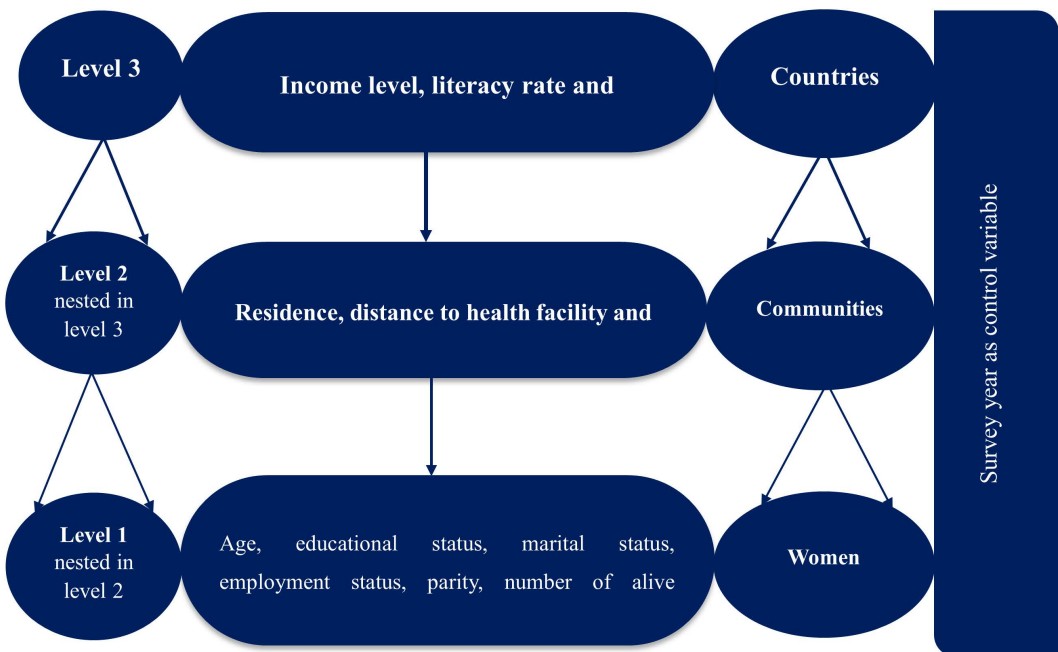

**Fig 1. Hierarchical nature of variables included in the study.**

sanitation facilities [19,20]. The HDI categorizes communities based on health, education, and standard of living [12,20]. Accordingly, communities are classified as having a high development index if they have access to electricity, improved sources of drinking water, and improved toilet facilities; otherwise, they are categorized as having a low development index.

**Country income level:** Country Income Levels: Country income levels are typically classified by organizations such as the World Bank, which categorizes countries into income groups based on Gross National Income (GNI) per capita [21]. The classifications include low-income, lower-middle-income, and upper-middle-income countries. This classification is updated annually and reflects the economic status of countries [21]. Researchers cross-referenced these classifications with the year of the most recent DHS data.

**Literacy rate:** Literacy rates are commonly categorized based on the percentage of the population that can read and write. Countries are assessed according to their literacy rates to determine educational development levels, which are linked to overall human development and economic performance [22,23]. We classified literacy rates as low (less than 75%), moderate (75–90%), and high (90% or more). This classification aligns with how various organizations, including UNESCO, report literacy statistics [22,24].

## Spatial analysis

**Spatial autocorrelation.** The spatial dependency of ANC service non-utilization in low- and middle-income countries was evaluated using Global Moran's I, which measures spatial autocorrelation. This statistic ranges from −1–1, with a value of 0 representing a random distribution, a value near −1 indicating a dispersed pattern, and a value close to 1 reflecting clustering. A statistically significant Moran's I value ($P < 0.05$) suggests the existence of spatial dependence [25].

**Hot spot analysis.** This study utilized an optimized hot spot analysis technique to identify regions with varying levels of ANC service non-utilization in low- and middle-income countries. The Optimized Hot Spot Analysis tool employs the Getis-Ord Gi* statistic, which is an enhanced version of the conventional Hot Spot Analysis. This tool detects statistically significant hot and cold spots in the data while controlling for multiple testing and spatial dependence through the False Discovery Rate (FDR) correction method [26]. The Getis-Ord Gi* statistic assesses spatial clustering by examining the distribution of features and their adjacent counterparts [26,27].

**Spatial interpolation.** The spatial prediction of non-utilization of ANC services in unsampled areas of the countries was conducted using Kriging interpolation, based on observed data from sampled regions. Kriging is an interpolation method that estimates the value of a variable in unsampled locations by utilizing observations from neighboring areas [28–30]. This technique minimizes prediction errors and represents geographic variation through a variogram. Named after Danie Krige, Kriging originated in the field of mining geology [29,30].

## Hierarchical analysis

For identifying the factors affecting the non-utilization of ANC services among pregnant women, we utilized data from the DHS and implemented a three-level hierarchical logistic regression model. This method considers the hierarchical nature of the data—encompassing individual, community, and country levels—enabling us to assess the impact of factors at each level while accounting for country-specific variations. We conducted a likelihood ratio test to compare this multilevel model with a standard logistic regression model, which demonstrated a notable enhancement in model fit with the multilevel approach, thereby affirming its appropriateness for our analysis.

In our examination of various factors, we developed five mixed-effect logistic regression models: a null model (Model 0) without explanatory variables, a model incorporating individual-level factors along with a control variable (Model I), a model including community-level factors and a control variable (Model II), a model that encompassed country-level factors with a control variable (Model III), and a comprehensive model (Model IV) that integrated all levels. We subsequently evaluated the random effects (which measure variability) and fixed effects (which assess associations) of each model.

Random effects were measured through variance, intra-class correlation coefficient (ICC), median odds ratio (MOR), and proportional change in variation (PCV), while fixed effects were expressed as adjusted odds ratios with 95% confidence intervals. Statistical significance was established using a p-value cutoff of 0.05. The three-level hierarchical regression model can be represented as follows [31,32]: $y_{ijk} = \beta 0k + \beta 1k x_{ijk} + r_{ijk} + u_{ijk}$

Where $y_{ijk}$ is the dependent variable for the i<sup>th</sup> individual in the j<sup>th</sup> community in the k<sup>th</sup> country, β0k and β1k are the intercept and slope coefficients for the k<sup>th</sup> country, $x_{ijk}$ is the independent variable for the i<sup>th</sup> individual in the j<sup>th</sup> community in the k<sup>th</sup> country, $r_{ijk}$ is the random effect for the j<sup>th</sup> community in the k<sup>th</sup> country, $u_{ijk}$ is the random effect for the i<sup>th</sup> individual in the j<sup>th</sup> community in the k<sup>th</sup> country.

Model selection involved a comparison of the log likelihood (LL), deviance (−2LL), and Bayesian Information Criterion (BIC) values across all models. Based on these metrics, Model IV was identified as the best-fitting model.

To validate the robustness of our results, we performed several checks. We evaluated multicollinearity—where independent variables may be closely related—by calculating the variance inflation factor (VIF) for each variable. The VIF values remained within acceptable limits, suggesting that multicollinearity was not a significant issue. Furthermore, we utilized multivariable regression techniques to account for potential confounding factors that could affect the relationship between the independent and dependent variables. This method enabled us to discern the effect of each independent variable while considering the influence of other pertinent factors.

### Ethical approval

This study was based on analysis of existing survey datasets in the public domain that are freely available online with all the identifier information anonymized, no ethical approval was required. The first author was obtained authorization for the download and usage of the DHS dataset of all countries included in the analysis from MEASURE DHS.

## Results

### Descriptive statistics of the study

The descriptive statistics shown in the Table 2 below provides an in-depth analysis of the non-utilization of ANC services across various characteristics in low- and middle-income countries. At the individual level, age-related trends reveal that the highest percentage of non-utilization occurs among women aged 35–49 (13.53%), while those aged 20–34 exhibit the lowest non-utilization rate (9.81%). Educational status significantly influences ANC utilization, with individuals lacking formal education demonstrating the highest non-utilization (22.20%) compared to those with higher education (96.36% utilized). Marital status also plays a role; women in union show a slightly higher non-utilization rate (10.68%) than their counterparts not in union (9.54%). Further, primipara women have lower non-utilization rates (11.95%) than multipara women (7.06%). Employment status reveals that employed individuals exhibit higher non-utilization (14.42%) compared to the unemployed (9.39%). Notably, non-utilization increases with the number of living children, with those having five or more children showing the highest non-utilization rate (18.92%). Additionally, households headed by males report higher non-utilization (10.92%) than those headed by females (9.09%). The wealth index indicates that the poorest individuals have the highest non-utilization rate (17.29%), while the richest have the lowest (5.40% non-utilized). Media exposure significantly affects utilization, with individuals lacking media access showing higher non-utilization (18.61%) compared to those with access (7.40%).

At the community level, residence type matters, with urban residents showing lower non-utilization (12.54) than rural residents (6.76%). Distance to health facilities also influences utilization, as those perceiving distance as a significant barrier have higher non-utilization rates (16.32%) compared to those who do not (8.14%). A lower community development index correlates with higher non-utilization (10.83% in low development areas).

At the country level, income levels reveal that low-income countries exhibit the highest non-utilization (13.97%), while upper-middle-income countries show a slightly lower rate (10.58%). Literacy rates further impact utilization; a low literacy

**Table 2. Descriptive statistics of non-utilization of ANC services by individual, community and county level characteristics (n = 480,068).**

| Characteristics | | Non-utilization of ANC services | | P value |
|---|---|---|---|---|
| | | Yes [N (%)] | No [N (%)] | |
| **Individual level characteristics** | | | | |
| Age | 13–19 | 2,966 (11.58) | 22,654 (88.42) | <0.001 |
| | 20–34 | 35,925 (9.81) | 330,288 (90.19) | |
| | 35–49 | 11,940(13.53) | 76,295 (86.47) | |
| Educational status | No education | 28,729 (22.20) | 100,700 (77.80) | <0.001 |
| | Primary school | 9,689 (8.87) | 99,569 (91.13) | |
| | Secondary school | 10,399 (5.59) | 175,715 (94.41) | |
| | Higher education | 2,013 (3.64) | 53,253 (96.36) | |
| Marital status | In union | 47051 (10.68) | 393394 (89.32) | <0.001 |
| | No in union | 3780 (9.54) | 35843 (90.46) | |
| Parity | Primipara | 9448 (7.06) | 124,412 (92.94) | <0.001 |
| | Multipara | 41383 (11.95) | 304825 (88.05) | |
| Employment status | Employed | 24572 (14.42) | 145855 (85.58) | <0.001 |
| | Unemployed | 15133 (9.39) | 146052 (90.61) | |
| Number of alive children | 1–2 | 21,874 (7.82) | 257,860 (92.18) | <0.001 |
| | 3–4 | 15,563 (12.01) | 113,983 (87.99) | |
| | 5 and more | 13,394 (18.92) | 57,395 (81.08) | |
| Sex of household head | Male | 42874 (10.92) | 349694 (89.08) | <0.001 |
| | Female | 7,957 9.09) | 79,543 (90.91) | |
| Wealth index | Poorest | 18320 (17.29) | 87641 (82.71) | <0.001 |
| | Poorer | 12417 (12.31) | 88460 (87.69) | |
| | Middle | 8774 (9.17) | 86897 (90.83) | |
| | Richer | 6761 (7.26) | 86410 (92.74) | |
| | Richest | 4559 (5.40) | 79829 (94.60) | |
| Media exposure | Yes | 24,366 (7.40) | 304,717 (92.60) | <0.001 |
| | No | 25935 (18.61) | 113412 (81.39) | |
| **Community level characteristics** | | | | |
| Residence | Urban | 10950 (6.76) | 150974 (93.24) | <0.001 |
| | Rural | 39881 (12.54) | 278263 (87.46) | |
| Distance to health facility | Big problem | 24,868 (16.32) | 127,552 (83.68) | <0.001 |
| | Not big problem | 25,112 (8.14) | 283,407 (91.86) | |
| Community development index | Low | 49,540 (10.83) | 407,841 (89.17) | |
| | High | 1,283 (5.66) | 21,379(94.34 | |
| **Country level characteristics** | | | | |
| Income level | Low | 19,465(13.97) | 119,863 (86.03) | <0.001 |
| | Lower-middle | 27,995(9.06) | 280,885 (90.94) | |
| | Upper-middle | 3,371 (10.58) | 28,490 (89.42) | |
| Literacy rate | Low | 30365 (18.41) | 134541 (81.59) | <0.001 |
| | Moderate | 17931 (6.70) | 249660 (93.30) | |
| | High | 2535 (5.33) | 45037 (94.67) | |
| Geographical region | Africa | 23147 (11.26) | 182451 (88.74) | <0.001 |
| | Asia | 24855 (9.89) | 226529 (90.11) | |
| | Europe | 290 (13.23) | 1901 (86.77) | |
| | Latine American & the Caribbean | 803 (5.63) | 13457 (94.37) | |
| | Oceania | 1737(26.17) | 4899 (73.83) | |

*(Continued)*

**Table 2.** (Continued)

| Characteristics | | Non-utilization of ANC services | | P value |
|---|---|---|---|---|
| | | Yes [N (%)] | No [N (%)] | |
| Survey year | 2015-2018 | 30029 (15.25) | 166836 (84.75) | <0.001 |
| | 2019-2024 | 20802 (7.35) | 262401 (92.65) | |

ANC: Antenatal care

rate corresponds to higher non-utilization (18.41%). Geographically, non-utilization rates vary, with Oceania showing the most significant service utilization (26.17% utilized) and Africa reflecting substantial non-utilization (11.26%). Survey years indicate notable differences, with the 2015–2018 period demonstrating higher non-utilization (15.25%) compared to the 2019–2024 period (7.35%). The P-values included in the table suggest that all of these chi-squared associations are statistically significant (P < 0.001), emphasizing the multifaceted nature of factors influencing ANC service utilization. This comprehensive overview highlights critical disparities related to individual, community, and country-level characteristics, underscoring the need for targeted interventions to improve ANC service uptake (Table 2).

## Magnitude of non-utilization of ANC services in low- and middle-income countries

The analysis of this study revealed that the overall magnitude of non-utilization of ANC services among women in low and middle-income countries was estimated at 10.59%, with a confidence interval ranging from 10.50% to 10.68%. The non-utilization rate was highest in Afghanistan at 40.05% and lowest in Burundi at 0.76% (Fig 2).

## Spatial distribution of non-utilization of ANC services in low- and middle-income countries

**Spatial autocorrelation of non-utilization of ANC services in low- and middle-income countries.** The spatial autocorrelation analysis of Non-Utilization of ANC services in LMICs revealed notable spatial patterns in service usage. The analysis yielded a Moran's Index of 0.098140, indicating a positive correlation in the distribution of non-utilization rates across these countries. The Z-score of 12.753093 reinforces this finding, as it significantly exceeds the threshold for randomness, and the associated p-value of less than 0.001 suggests that the observed clustering of non-utilization is highly unlikely to occur by chance. This strong statistical evidence indicates that areas with low utilization of ANC services tend to be clustered rather than randomly distributed, highlighting the need for targeted interventions in specific regions. The accompanying figure (Fig 3) visually illustrates these patterns, with distinct areas of clustering. Overall, this analysis underscores the importance of considering spatial factors when addressing the disparities in ANC service utilization across LMICs.

**Hotspot analysis of non-utilization of ANC services in low- and middle-income countries.** Fig 4 illustrates the geographic distribution of non-utilization of ANC services among women in low- and middle-income countries. It identifies both hot spots and cold spots, providing insights into regional disparities in utilization of ANC services. The hot spots are regions marked in red and orange, indicating areas with a significantly higher likelihood of non-utilization of ANC services. Notably, countries and regions within South Asia and parts of Sub-Saharan Africa exhibit a high concentration of hot spots, suggesting systemic barriers that prevent women from accessing essential prenatal care. Specifically, countries such as Albania, Angola, Ethiopia, Guinea, East India, Madagascar, Mali, Mauritania, Nepal, Nigeria, and East Timor: Factors contributing to this trend may include socio-economic challenges, cultural beliefs, inadequate healthcare infrastructure, and limited availability of healthcare providers.

In contrast, the cold spots, represented in green, indicate regions where the non-utilization of ANC services is significantly lower. These areas demonstrate better maternal health outcomes and are often linked to higher levels of education,

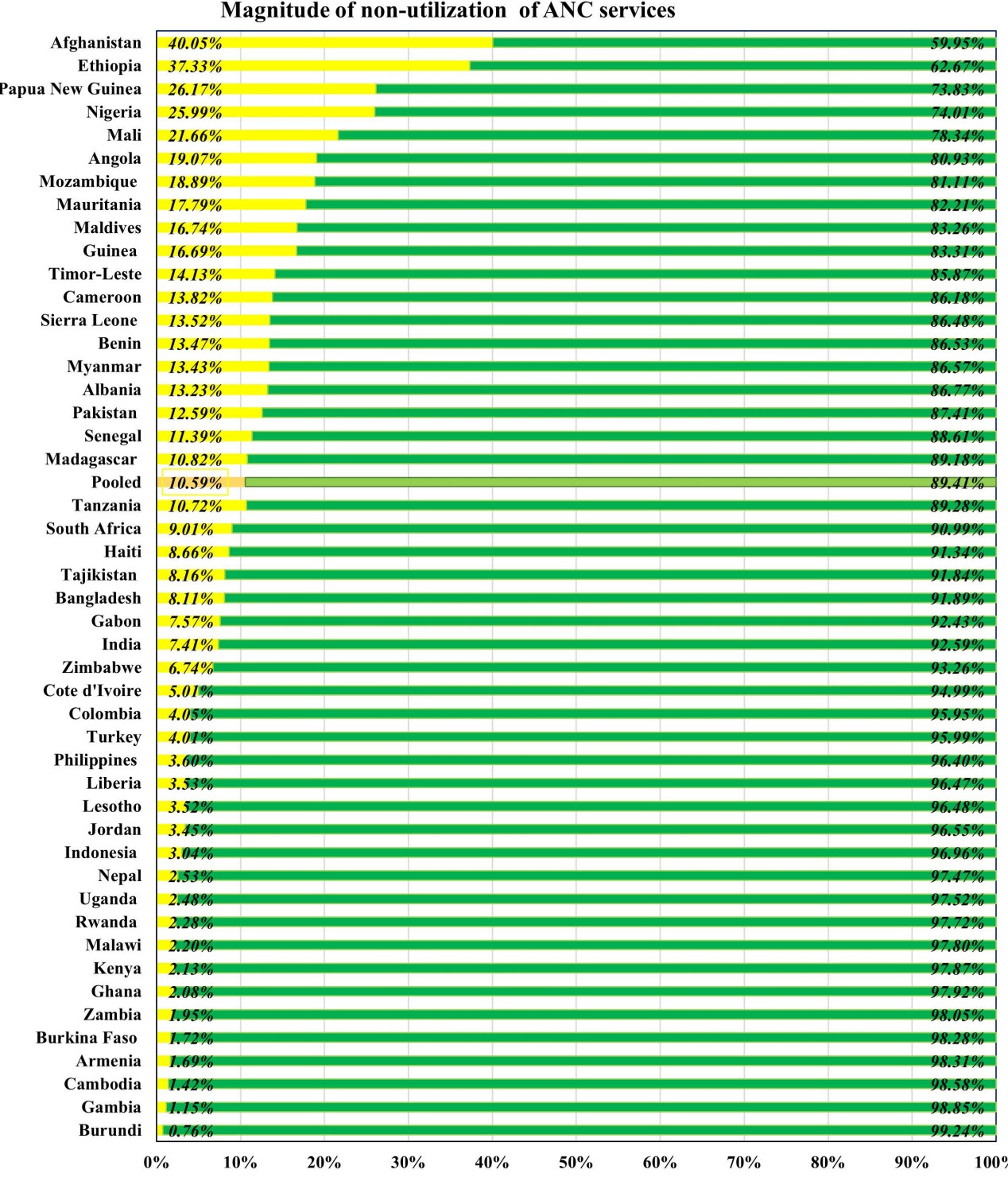

**Fig 2. The pooled and national magnitude of non-utilization of ANC services.**

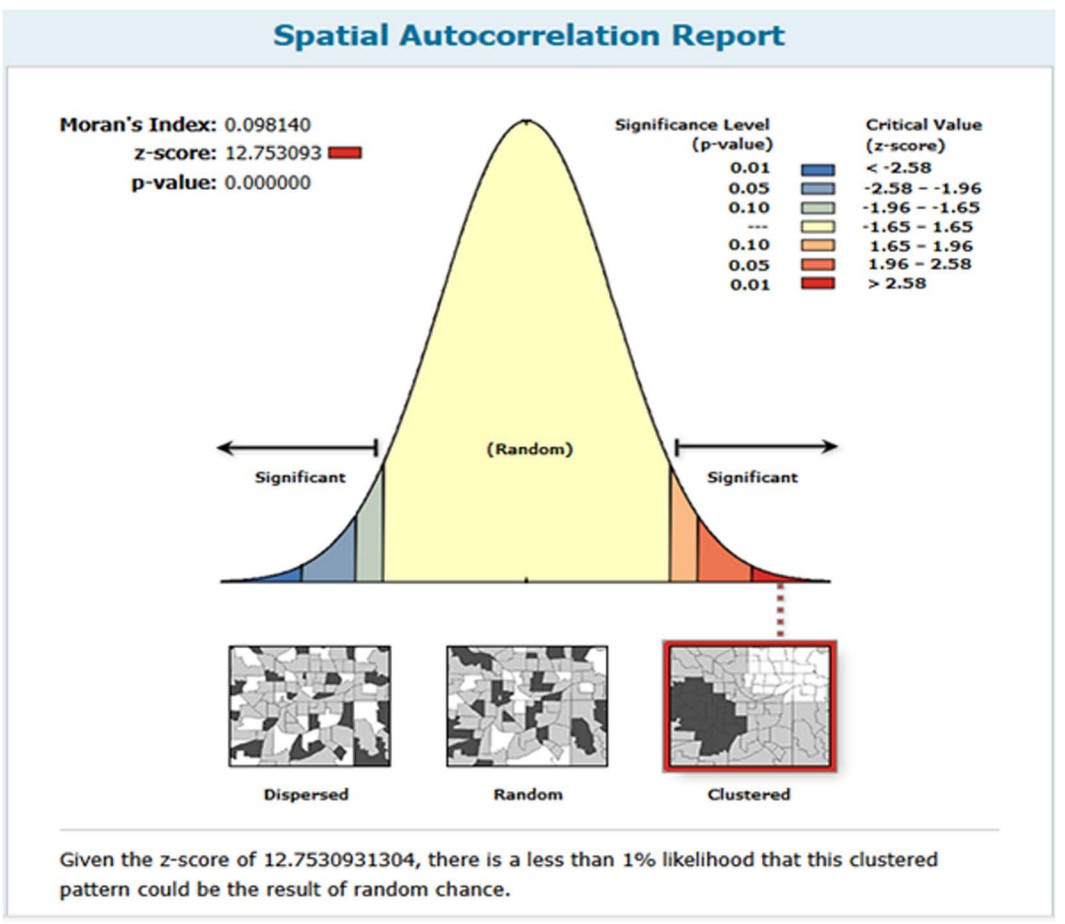

**Fig 3. Spatial autocorrelation of non-utilization of ANC services in low- and middle-income countries.**

economic stability, and effective healthcare systems. Countries such as Burundi, Haiti, Jordan, East of Ghana, Kenya, the Philippines, Rwanda and Tajikistan show a prevalence of cold spots, reflecting effective public health strategies and increased awareness of the importance of antenatal care (Fig 4).

**Interpolation of non-utilization of ANC services in low- and middle-income countries.** The result of the Kriging interpolation is illustrated in Fig 5, showcasing the predicted values for the non-utilization of ANC services among women in low- and middle-income countries. The color gradient ranges from low values represented in green to high values depicted in yellow and orange. The areas highlighted in orange indicate regions with a higher predicted rate of non-utilization of ANC services, suggesting significant barriers to accessing these essential healthcare services. These hotspots are primarily concentrated in specific geographical locations, including most parts of Angola, eastern parts of Ethiopia, west of Pakistan, east of Mali, and some portions of eastern Myanmar. Conversely, regions shaded in green represent areas with lower predicted rates of non-utilization, indicating better access to ANC services. These areas tend to have relatively stronger healthcare systems, higher educational levels, and greater community awareness about the importance of antenatal care. The predicted value of non-utilization of ANC services ranges from a high of 1.0514 to a low of −0.185949, reflecting a significant disparity in access and utilization across different regions (Fig 5).

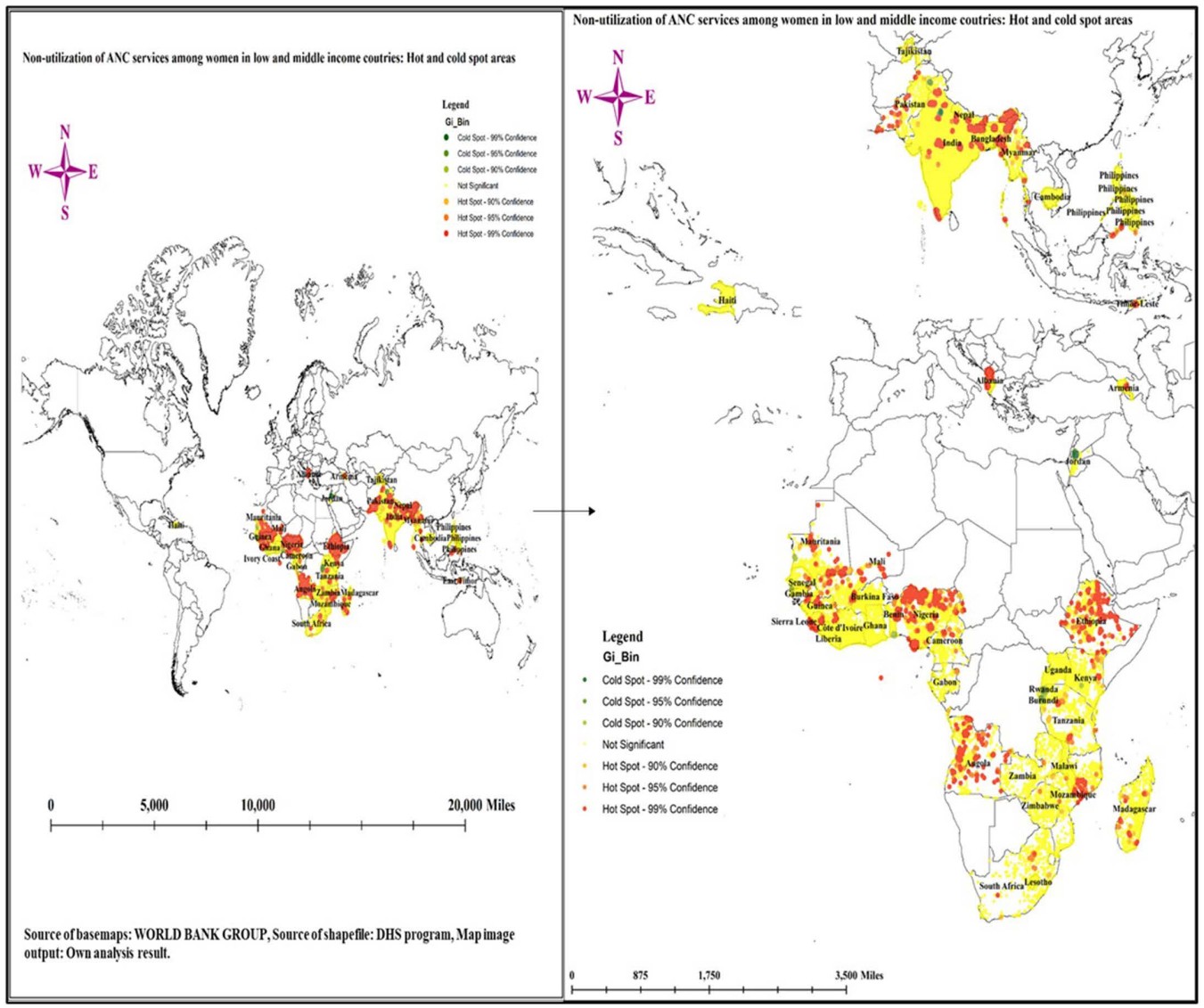

**Fig 4. Hot spot analysis of non-utilization of ANC services in low- and middle-income countries; Source basemaps URL:** https://datacatalog. worldbank.org/search/dataset/0038272/World-Bank-Official-Boundaries); **the figure is similar but not identical to the original image and is therefore for illustrative purposes only.**

### Hierarchical analysis of non-utilization of utilization of ANC services

**Model fitness.** The model fitness metrics of this study was presented in Table 3 for five different models, including a null model and four specified models (Model I to Model IV). The log-likelihood values indicate how well each model fits the data, with the highest value for Model IV at −92,796.44, suggesting it provides the best fit among the models evaluated. The deviance metric, which measures the goodness of fit, is lowest for Model IV at 185,592.88, indicating a superior fit compared to the others. In terms of the AIC, which penalizes for model complexity, Model IV again shows the best value at 185,650.90. Lastly, the BIC also favors Model IV with a score of 185,960.10, reinforcing its status as the most effective model among those considered. Overall, the metrices clearly indicate that Model IV outperforms the other models in terms of fit and complexity.

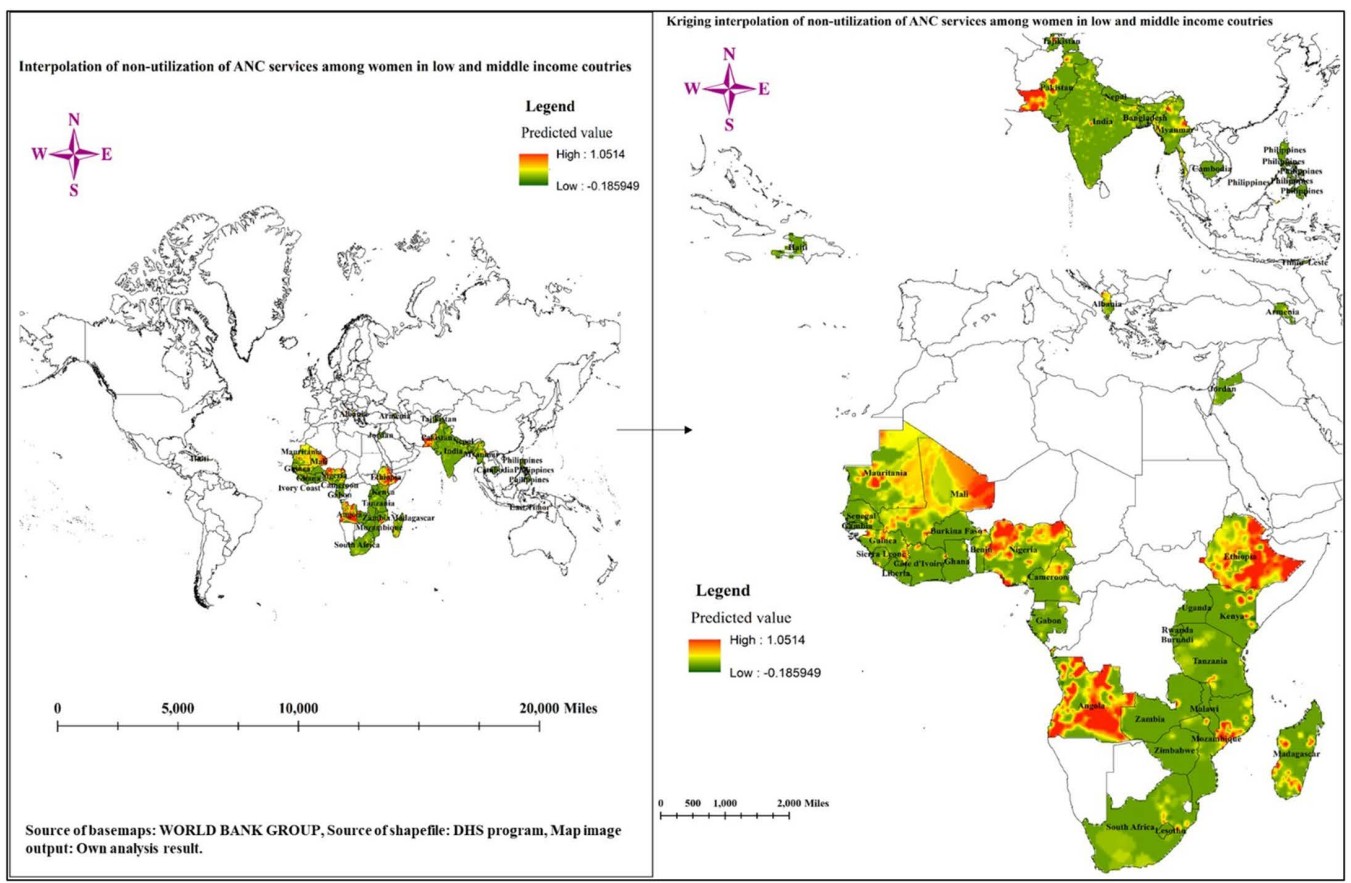

**Fig 5. Interpolation of non-utilization of ANC services in low- and middle-income countries; Source basemaps URL:** https://datacatalog.world-bank.org/search/dataset/0038272/World-Bank-Official-Boundaries**); the figure is similar but not identical to the original image and is therefore for illustrative purposes only.**

**Fixed effects.** The fixed effects section of our analysis shows that a total of seven factors were significantly associated with the non-utilization of ANC services among women during pregnancy. Specifically, these factors included educational level, marital status, household wealth index, media exposure, residence, distance to health facility and income level of the countries.

The odds of non-utilization of ANC services were 3.28, 1.81, and 1.28 times higher among women with no education (AOR = 3.28; 95% CI: 3.02–3.55), primary schooling (AOR = 1.81; 95% CI: 1.67–1.96), and secondary schooling (AOR = 1.28; 95% CI: 1.18–1.38), respectively, compared to women with higher educational status.

In terms of marital status, the odds of non-utilization of ANC services were 1.35 times higher among women not in union compared to those in union. Moreover, the odds of non-utilization of ANC services were 1.87, 1.45, 1.17, and 1.09 times higher among women in the poorest (AOR = 1.87; 95% CI: 1.77–1.98), poorer (AOR = 1.45; 95% CI: 1.38–1.54), middle (AOR = 1.17; 95% CI: 1.11–1.24), and richer wealth index categories (AOR = 1.09; 95% CI: 1.04–1.15), respectively, compared to women living in households with the richest wealth index.

The likelihood of non-utilization of ANC services was 1.68 times higher among women with no media exposure (AOR = 1.68; 95% CI: 1.64–1.73) compared to those with media exposure. Additionally, the odds of non-utilization of ANC were 5% higher among women who were rural residents (AOR = 1.05; 95% CI: 1.02–1.09) compared to women living in

**Table 3. Hierarchical analysis of non-utilization of utilization of ANC services in low- and middle-income countries, DHS 2015-2024.**

| Model fitness | | | | | |
|---|---|---|---|---|---|
| **Metrics** | **Null model** | **Model I** | **Model II** | **Model III** | **Model IV** |
| Log likelihood | −149288.92 | −100404.53 | −140267.67 | −143329.76 | −92796.44 |
| Deviance | 298,577.84 | 200,809.06 | 280,535.34 | 286,659.52 | 185,592.88 |
| AIC | 298581.80 | 200847.10 | 280547.30 | 286681.5 | 185650.90 |
| BIC | 298604.00 | 201050.10 | 280613.70 | 286803.6 | 185960.10 |
| **Fixed effect** | | | | | |
| **Factors** | | **Model I AOR with 95% CI** | **Model II AOR with 95% CI** | **Model III AOR with 95% CI** | **Model IV AOR with 95% CI** |
| Survey year | 2015-2018 | 1.80 (1.10, 3.15) | 2.16 (2.10, 2.22) | 1.75 (1.00, 3.02) | 1.72 (0.98, 2.99) |
| | 2019-2024 | 1.00 | 1.00 | 1.00 | 1.00 |
| Age | 13-19 | 1.08 (0.93, 1.25) | | | 1.06 (0.90, 1.23) |
| | 20-34 | 1.00 | | | 1.00 |
| | 35-49 | 0.99 (0.95, 1.02) | | | 0.97 (0.94, 1.00) |
| Educational status | No education | 3.30 (3.05, 3.57) | | | **3.28 (3.02, 3.55) *** |
| | Primary | 1.85 (1.69, 1.97) | | | **1.81 (1.67, 1.96) *** |
| | Secondary | 1.29 (1.19, 1.40) | | | **1.28 (1.18, 1.38) *** |
| | Higher | 1.00 | | | 1.00 |
| Marital status | In union | 1.00 | | | 1.00 |
| | No in union | 1.36 (1.30, 1.43) | | | **1.35 (1.29, 1.41) *** |
| Parity | Primipara | 1.00 | | | 1.00 |
| | Multipara | 1.11 (0.95, 1.25) | | | 1.09 (0.93, 1.23) |
| Employment status | Employed | 1.00 | | | 1.00 |
| | Unemployed | 1.09 (0.96, 1.34) | | | 1.08 (0.95, 1.32) |
| Number of alive children | 1-2 | 1.00 | | | 1.00 |
| | 3-4 | 1.10 (0.87, 1.14) | | | 1.08 (0.85, 1.12) |
| | 5 and more | 1.03 (0.97, 1.26) | | | 0.92 (0.76, 1.25) |
| Sex of household head | Male | 1.05 (0.99, 1.08) | | | 1.03 (0.98, 1.07) |
| | Female | 1.00 | | | 1.00 |
| Wealth index | Poorest | 1.89 (1.77, 1.98) | | | **1.87 (1.77, 1.98) *** |
| | Poorer | 1.47 (1.38, 1.54) | | | **1.45 (1.38, 1.54) *** |
| | Middle | 1.19 (1.13, 1.25) | | | **1.17 (1.11, 1.24) *** |
| | Richer | 1.10 (1.05, 1.16) | | | **1.09 (1.04, 1.15) *** |
| | Richest | 1.00 | | | 1.00 |
| Media exposure | Yes | 1.00 | | | 1.00 |
| | No | 1.69 (1.65, 1.75) | | | **1.68 (1.64, 1.73) *** |
| Residence | Urban | | 1.00 | | 1.00 |
| | Rural | | 1.09 (1.06, 1.11) | | **1.05 (1.02, 1.09) *** |
| Distance to HF | Big problem | | 1.35 (1.30, 1.37) | | **1.31 (1.28, 1.34) *** |
| | Not big problem | | 1.00 | | 1.00 |
| CDI | Low | | 1.00 (0.94, 1.09) | | 0.99 (0.92, 1.07) |
| | High | | 1.00 | | 1.00 |
| Income level | Low | | | 2.28 (1.25, 8.75) | **2.27 (1.23, 6.74) *** |
| | Lower-middle | | | 0.42 (0.17, 1.09) | 0.40 (0.15, 1.08) |
| | Upper-middle | | | 1.00 | 1.00 |

*(Continued)*

**Table 3.** (Continued)

| Model fitness | | | | | | |
|---|---|---|---|---|---|---|
| **Metrics** | | **Null model** | **Model I** | **Model II** | **Model III** | **Model IV** |
| Literacy rate | Low | | | | 1.69 (0.74, 3.87) | 1.68 (0.73, 3.86) |
| | Moderate | | | | 0.80 (0.36, 1.93) | 0.82 (0.34, 1.95) |
| | High | | | | 1.00 | 1.00 |
| Geographical region | Africa | | | | 0.94 (0.18, 5.25) | 0.92 (0.16, 5.22) |
| | Asia | | | | 1.19 (0.21, 7.09) | 1.16 (0.19, 7.06) |
| | Europe | | | | 1.30(0.13, 16.61) | 1.27 (0.10, 16.58) |
| | LAC | | | | 1.00 | 1.00 |
| | Oceania | | | | 2.00 (0.22, 2.25) | 1.97 (0.18. 2.20) |
| **Radom effect** | | | | | | |
| **Metrics** | | **Null model** | **Model I** | **Model II** | **Model III** | **Model IV** |
| Variance | Country | 2.34(1.68, 3.39) | 0.72 (0.45, 1.06) | 2.18 (1.55, 3.15) | 0.76 (0.51, 1.15) | 0.65 (0.47, 1.08) |
| | Community | 2.84 (2.71, 2.98) | 0.87 (0.78, 0.98) | 2.64 (2.51, 2.77) | 2.63 (2.51, 2.76) | 0.85 (0.76, 0.95) |
| ICC (%) | Country | 38.17 (27.98, 54.06) | 17.76 (12.45, 24.69) | 37.73 (28.09, 49.43) | 18.81 (13.35, 25.85) | 17.74 (12.43, 24.66) |
| | Community | 46.33 (45.14, 47.52) | 20.95 (19.14, 22.88) | 44.52 (43.32, 45.73) | 44.45 (43.27, 45.63) | 20.93 (19.17, 22.81) |
| MOR | Country | 4.28 (3.43, 5.75) | 2.24 (1.89, 2.66) | 4.06 (3.26, 5.40) | 2.29 (1.97, 2.77) | 2.23 (1.92, 2.68) |
| | Community | 4.95 (4.77, 5.15) | 2.43 (2.31, 2.56) | 4.68 (4.50, 4.86) | 4.67 (2.59, 4.85) | 2.41 (2.29, 2.52) |
| PCV | Country | Reference | 69.23% | 6.84% | 67.52% | 72.22% |
| | Community | Reference | 69.37% | 7.04% | 7.39% | 70.07% |

AIC: Akaike Information Criterion, BIC: Bayesian Information Criterion, CDI: Community development index, HF: Health Facility, LAC: Latine American & the Caribbean

urban areas. Furthermore, perceiving distance to a health facility as a significant problem was associated with higher odds of non-utilization of ANC (AOR = 1.31; 95% CI: 1.28–1.34). Lastly, the likelihood of non-utilization of ANC services was 2.27 times higher among women from countries with low-income levels (AOR = 2.27; 95% CI: 1.23–6.74) compared to those residing in upper-middle-income countries (Table 3).

**Random effects.** The random effects section of Table 3 presents a detailed examination of variance components, intraclass correlation coefficients, median odds ratio, and proportional change in variance across various models, highlighting the influence of country and community-level factors on the non-utilization of ANC services, in addition to individual-level variables.

The variance attributed to the country level starts at 2.34 (95% CI: 1.68, 3.39) in the null model, indicating substantial variation among countries regarding ANC service utilization. This variance decreases significantly to 0.72 (95% CI: 0.45, 1.06) in Model I, suggesting that the inclusion of certain covariates has explained a considerable portion of the variance. Subsequent models continue this trend, with Model IV showing a variance of 0.65 (95% CI: 0.47, 1.08). In contrast, the community-level variance begins at 2.84 (95% CI: 2.71, 2.98) and declines to 0.87 (95% CI: 0.78, 0.98) in Model I, stabilizing at 0.85 (95% CI: 0.76, 0.95) in Model IV. This reduction indicates that community-level characteristics play a significant role in explaining variations in ANC service utilization.

The ICC serves as a measure of the proportion of total variance attributable to group-level differences. For the country level, the ICC starts at 38.17% (95% CI: 27.98, 54.06) in the null model, reflecting that a substantial portion of the variance in ANC non-utilization can be attributed to differences between countries. This value decreases to 17.76% (95% CI: 12.45, 24.69) in Model I, indicating that the covariates included in this model account for a significant amount of the country-level

variance. The community-level ICC follows a similar pattern, beginning at 46.33% (95% CI: 45.14, 47.52) and dropping to 20.95% (95% CI: 19.14, 22.88) in Model I. These trends suggest that both country and community-level factors significantly contribute to understanding ANC service utilization.

The Median Odds Ratio quantifies the odds of non-utilization of ANC services between two randomly selected individuals from different groups (different countries or communities), providing insight into the degree of heterogeneity in non-utilization rates due to group-level factors. In the null model, the MOR for country-level factors is 4.28 (95% CI: 3.43, 5.75), indicating that a randomly selected individual from one country has approximately 4.28 times the odds of non-utilization compared to a randomly selected individual from another country. This substantial MOR reflects significant disparities in ANC service utilization across countries. As we progress through the models, the MOR for country-level factors decreases to 2.24 (95% CI: 1.89, 2.66) in Model I, suggesting that the introduction of covariates reduces the odds of non-utilization between individuals in different countries, thereby accounting for some of the between-country variability. Similarly, the community-level MOR starts at 4.95 (95% CI: 4.77, 5.15) in the null model, highlighting notable differences in non-utilization rates at the community level. In Model I, the MOR decreases to 2.43 (95% CI: 2.31, 2.56), reflecting that the inclusion of community-level characteristics explains some variability in ANC non-utilization. The decreasing trend in MOR values across models emphasizes that while significant disparities exist in ANC service utilization at both country and community levels, these disparities can be partially accounted for by the characteristics considered in the analysis. Thus, the MOR serves as a crucial indicator for understanding how much of the variation in non-utilization can be attributed to contextual factors, guiding targeted interventions to improve ANC service uptake in specific populations.

The PCV indicates the proportion of variance explained by the models in comparison to the null model. At the country level, the PCV increases from the null model (reference) to Model II (6.84%) and reaches 72.22% in Model IV, demonstrating that the final model more effectively explains a high proportion of the variance in ANC non-utilization. Similarly, community-level PCV values show a substantial increase, starting from a reference point and culminating at 70.07% in Model IV. This indicates that country and community-level factors, in addition to individual-level variables, are crucial in explaining variations in ANC service utilization.

Therefore, the random effects results underscore the importance of both country and community-level factors in understanding the non-utilization of ANC services, in addition to the individual-level factors, as evidenced by the substantial reductions in variance and changes in ICC, MOR, and PCV across the models (Table 3).

## Discussion

Antenatal care is a crucial component of reproductive health care, offering a platform for health promotion, screening, diagnosis, and disease prevention [2,3]. Evidence-based practices implemented in a timely and appropriate manner through ANC can save lives [4]. Despite significant advancements in expanding access to health services in LMICs, the quality and access of care provided varies greatly across different countries and health conditions [12,13]. This inconsistency hampers progress in enhancing health outcomes [13]. This study assessed the magnitude, geographic distribution, and determinants of non-utilization of ANC services in LMICs, using data from the most recent DHSs conducted between 2015 and 2024.

This study revealed that the overall magnitude of non-utilization of ANC services among women in low and middle-income countries was estimated at 10.59%, with a confidence interval ranging from 10.50% to 10.68%. This statistic aligns with broader trends observed in the literature, where significant proportions of women in LMICs fail to meet the WHO's recommendation of a minimum of eight ANC visits during pregnancy [8,33,34]. Despite ongoing efforts to promote ANC services, many studies corroborate our findings, revealing that a substantial number of women—especially those in rural and underserved areas—are not receiving adequate prenatal care [33,34].

Interestingly, the magnitude of non-utilization of ANC services identified in our study is notably lower than a previous study conducted in South Sudan, which reported a non-utilization rate of 58% [35]. This discrepancy may be attributed to

differences in study design, methodologies, and the temporal context of the research. More recent studies, including ours, may reflect improvements in healthcare access and awareness, leading to higher utilization rates. As access to information and healthcare services increases, women are more likely to seek and utilize ANC services, thereby enhancing maternal and child health outcomes.

The study revealed a striking disparity in the non-utilization rates across LMICs, ranging from as low as 0.76% in Burundi to as high as 40.05% in Afghanistan. These figures go beyond statistical variation—they reflect deep-seated structural, cultural, and policy-related differences that shape maternal health service uptake. In countries such as Burundi, the near-universal use of ANC services (99.24% utilization) was likely driven by strong investments in community-based health systems, where health workers engage directly with households to ensure early and continued maternal care. Burundi's success has been linked to its robust health extension program and results-based financing mechanisms that prioritize maternal and child health [36]. These strategies have helped bridge geographic and economic barriers, ensuring even women in remote areas receive timely ANC. In contrast, Afghanistan, with a non-utilization rate of 40.05%, faces profound challenges. Prolonged conflict, poor infrastructure, political instability, and conservative gender norms have severely constrained women's access to maternal services [37,38]. Similarly, Ethiopia (37.33%) and Papua New Guinea (26.17%) exhibit high levels of ANC non-utilization. These settings are often marked by rugged geography, poor road networks, and insufficient health infrastructure, which disproportionately affect rural and indigenous populations [1,39]. Health literacy may also play a role; in regions where awareness of ANC's importance is low, utilization remains limited despite improvements in service availability [40]. This wide range in ANC non-utilization underscores the importance of context-sensitive interventions. In high-burden countries, efforts should address both supply-side constraints (inadequate facilities, staff shortages) and demand-side barriers such as cultural norms, lack of autonomy, poor education) [1,37–39].

The Spatial analysis conducted in this study reveals critical insights into the geographic distribution of non-utilization of ANC services among women in LMICs. The identification of both hot spots and cold spots highlights significant regional disparities in ANC service utilization, with hot spots represented in red and orange, indicating areas with a markedly higher likelihood of non-utilization. Notably, regions within South Asia and parts of Sub-Saharan Africa demonstrate a high concentration of these hot spots, suggesting systemic barriers that hinder women's access to essential prenatal care. Countries such as Albania, Angola, Ethiopia, Guinea, East India, Madagascar, Mali, Mauritania, Nepal, Nigeria, and East Timor are particularly affected. The highest magnitude of non-utilization of ANC services in countries such as Afghanistan, Ethiopia, Papua New Guinea, Nigeria, Mali, Angola, Mauritania, Maldives and Guinea support the finding from the spatial analysis. The factors contributing to this trend in these countries are multifaceted and may include socio-economic challenges, cultural beliefs, inadequate healthcare infrastructure, and limited availability of healthcare providers [25,41–43]. For instance, socio-economic barriers such as poverty and lack of education significantly impact women's ability to access ANC services, as evidenced by studies showing that women from poorer backgrounds are less likely to utilize these services [8]. Cultural beliefs and practices also play a crucial role; in many communities, traditional practices may overshadow the perceived necessity of formal healthcare, leading to lower utilization rates [44]. Moreover, inadequate healthcare infrastructure, particularly in rural areas, exacerbates the situation. Many women face logistical challenges, such as long distances to healthcare facilities and insufficient transportation options, which deter them from seeking ANC services [45]. This finding highlights the need for targeted interventions and policy measures in the identified high-risk areas to improve access to ANC services and ultimately enhance maternal and child health outcomes.

The hierarchical analysis of this study revealed that educational level, marital status, household wealth index, media exposure, residence, distance to health facility, and income level of the countries were significantly associated with the non-utilization of ANC services among women during pregnancy.

Specifically, the odds of non-utilization of ANC services were higher among women with no formal education or have low educational level compared to women with higher educational status. This finding aligns with previous studies [46–48]. The Health Belief Model also supports this relationship, suggesting that higher education increases perceived

susceptibility to health issues, thereby fostering proactive health-seeking behaviors [49]. Educated women generally have better access to information regarding the importance of ANC, allowing them to understand the risks associated with inadequate prenatal care. Furthermore, education empowers women to advocate for their health needs, as studies indicate that those with higher educational levels are more likely to engage in discussions with healthcare providers and actively seek ANC services [50]. Additionally, education is often linked to improved socioeconomic status, which directly influences healthcare access; women with higher education typically enjoy better job opportunities and increased financial resources that facilitate ANC utilization [48]. Education also plays a crucial role in shifting cultural perceptions regarding healthcare; in communities where education is valued, women may be more inclined to challenge norms that discourage seeking medical help [51]. Therefore, these findings underscore the importance of educational attainment in enhancing the utilization of ANC services, ultimately leading to improved maternal and child health outcomes.

The finding that women not in union have higher odds of non-utilization of ANC services compared to those who are married or in a union highlights the significant role that marital status plays in accessing healthcare. This finding is in line with literature [10,48]. Women in stable unions often benefit from increased support from partners, which can facilitate access to healthcare services. Research indicates that marital status can influence health-seeking behavior [52,53]. Women in unions are generally more empowered to seek healthcare due to shared responsibilities and resources with their partners [52]. They may also receive encouragement from their spouses to attend ANC appointments, which can lead to higher utilization rates. Conversely, women not in union may experience isolation or lack of support, making it more challenging to prioritize and access necessary healthcare services during pregnancy. Additionally, cultural norms and societal expectations often dictate that married women are more likely to engage in health-seeking behaviors [53]. In many societies, being in a union is associated with increased legitimacy and acceptance, which can translate into better access to healthcare resources. Unmarried women, on the other hand, may encounter barriers such as discrimination or lack of information about available services, further contributing to the lower rates of ANC utilization among this group [53].

The current study coherent with literature [54–57] found that the odds of non-utilization of ANC services were higher among women in the households with low wealth quintile, compared to women living in households with the highest wealth index. Firstly, financial barriers play a crucial role. Women in lower wealth quintiles often face direct costs such as consultation fees and transportation, as well as indirect costs like lost wages, which can deter them from seeking ANC services. Additionally, these women may lack health insurance or access to government health programs that could alleviate some of these financial burdens [54]. Secondly, access to information and education about the importance of ANC is typically better among wealthier households. Women in higher wealth quintiles are more likely to be educated and informed about the benefits of regular ANC visits, leading to higher utilization rates [57]. In contrast, women in lower wealth quintiles may not have the same level of awareness or understanding, which can result in lower utilization of ANC services. This is supported by a study in Mandera County, Kenya, which found that household wealth positively affected ANC service utilization [55]. Thirdly, the quality and availability of healthcare infrastructure are often better in wealthier areas. Women in higher wealth quintiles are more likely to live in urban areas with better healthcare facilities and services [56]. Conversely, women in lower wealth quintiles may reside in rural or underserved areas where healthcare facilities are scarce or of lower quality, further hindering their access to ANC services [57].

The likelihood of non-utilization of ANC services was higher among women with no media exposure compared to those with media exposure. Similar finding has also been reported by previous studies [58–60]. This finding underscores the critical role of media in promoting health-seeking behaviors. Media exposure significantly enhances awareness and knowledge about the importance of ANC services. Women who have access to media such as radio, television, and newspapers are more likely to be informed about the benefits of regular ANC visits, the services available, and the potential risks of not attending these visits [59]. Additionally, media campaigns can effectively influence health behaviors by disseminating information and promoting positive health practices [58]. Research in Uganda demonstrated that women who listened to the radio or watched television were more likely to initiate ANC earlier, which is crucial for timely identification

and management of pregnancy-related complications. This early initiation of ANC is associated with better maternal and neonatal outcomes [58]. Furthermore, media serves as a vital source of information, especially in areas where healthcare services and educational opportunities are limited. Women with media exposure are more likely to receive consistent and accurate information about maternal health, which can empower them to seek ANC services. This is particularly important in rural and underserved areas where other sources of information may be scarce. Media can also play a role in shaping social norms and providing social support for health-seeking behaviors [58]. By showcasing positive stories and testimonials about ANC, media can help normalize the practice and encourage women to utilize these services, which can be especially impactful in communities where there may be cultural or social barriers to seeking ANC [58,59]. Governments and health organizations can leverage media to support public health campaigns and policies aimed at increasing ANC utilization [58]. By using media to disseminate information about free or subsidized ANC services, vaccination programs, and other maternal health initiatives, policymakers can reach a broader audience and encourage higher utilization rates. Therefore, the positive impact of media exposure on ANC utilization underscores the need for targeted media campaigns and interventions to improve maternal health outcomes.

Additionally, the odds of non-utilization of ANC were higher among women who were rural residents compared to women living in urban areas. Prior studies have also witnessed the same [43,54,60,61]. It highlights the significant geographic disparities in healthcare access. This disparity could be attributed to several factors: Firstly, healthcare infrastructure in rural areas is often less developed compared to urban areas. Rural regions may have fewer healthcare facilities, limited availability of healthcare professionals, and inadequate medical supplies and equipment. This lack of infrastructure can make it difficult for rural women to access ANC services. A study in Ethiopia found that women in rural areas were less likely to receive ANC services due to the long distances to healthcare facilities and the poor quality of services available [61]. Secondly, transportation barriers are more pronounced in rural areas [60]. Women in rural regions may have to travel long distances on poor roads to reach healthcare facilities, which can be both time-consuming and costly. This can deter them from seeking ANC services, especially if they have to make multiple visits during their pregnancy [60]. Additionally, socioeconomic factors also play a role. Rural areas often have higher levels of poverty and lower levels of education compared to urban areas. Women in rural regions may lack the financial resources to afford transportation and healthcare costs, and they may have lower levels of awareness about the importance of ANC services. A study in Nigeria found that rural women were less likely to utilize ANC services due to financial constraints and lack of awareness [54]. Moreover, cultural and social factors can influence ANC utilization in rural areas. Traditional beliefs and practices may discourage women from seeking formal healthcare services, and there may be a preference for home-based care or traditional birth attendants. Social norms and gender roles in rural communities can also limit women's autonomy and decision-

making power regarding their health. A study in India found that cultural beliefs and gender norms were significant barriers to ANC utilization among rural women [43]. Hence, addressing the geographic disparities in ANC utilization requires targeted interventions to improve healthcare infrastructure, transportation, and socioeconomic conditions in rural areas. Policymakers and healthcare providers need to develop strategies to make ANC services more accessible and affordable for rural women, such as mobile health clinics, community health worker programs, and public awareness campaigns.

In this study, perceiving distance to a health facility as a significant problem was associated with higher odds of non-utilization of ANC. The finding underscores the critical impact of geographic accessibility on healthcare utilization. This issue can be attributed to several factors: Firstly, the physical distance to healthcare facilities can be a substantial barrier for many women, particularly those in rural or remote areas. Long distances often mean increased travel time and costs, which can deter women from seeking ANC services. A study conducted in Tanzania found that distance to healthcare facilities was a significant barrier to ANC utilization, with women living farther away being less likely to attend ANC visits.

In addition, the perception of distance as a barrier can be influenced by the quality of transportation infrastructure. Poor road conditions, lack of reliable public transportation, and high transportation costs can exacerbate the challenges

associated with accessing healthcare facilities [60–62]. A study in Ethiopia highlighted that women in rural areas often face long and difficult journeys to reach healthcare services, which significantly impacts their utilization of ANC [61].

Moreover, the availability of healthcare facilities within a reasonable distance is crucial. In many low- and middle-income countries, healthcare facilities are unevenly distributed, with urban areas having better access compared to rural regions. This disparity can lead to lower utilization of ANC services among women who faced the distance to the nearest facility as a significant obstacle. A multi-country study found that even relatively small distances from health facilities were associated with substantial reductions in healthcare utilization, including ANC services [62]. Therefore, addressing the issue of distance to healthcare facilities requires a multifaceted approach. Policymakers and healthcare providers need to improve the availability and accessibility of healthcare services, particularly in rural and underserved areas. This can include investing in better transportation infrastructure, establishing more healthcare facilities closer to communities, and implementing mobile health clinics. Additionally, public awareness campaigns and community engagement initiatives can help change perceptions and encourage women to seek ANC services despite the challenges posed by distance.

Lastly, the likelihood of non-utilization of ANC services was higher among women from countries with low-income levels compared to those residing in upper-middle-income countries. It highlights the significant impact of national income levels on healthcare access and utilization. This disparity can be attributed to the following reasons. Firstly, healthcare infrastructure and resources are generally more developed in upper-middle-income countries compared to low-income countries. Upper-middle-income countries typically have better-funded healthcare systems, more healthcare facilities, and a higher availability of trained healthcare professionals. This improved infrastructure facilitates greater access to ANC services for women [63]. Secondly, economic stability and higher income levels in upper-middle-income countries enable better healthcare financing and insurance coverage [63,64]. Women in these countries are more likely to have health insurance or access to government-funded healthcare programs that reduce the financial burden of ANC services. In contrast, women in low-income countries often face out-of-pocket expenses for healthcare, which can be a significant barrier to utilizing ANC services [64]. Additionally, education and awareness about the importance of ANC are generally higher in upper-middle-income countries. Higher levels of education and better access to information through media and public health campaigns contribute to greater awareness and understanding of the benefits of ANC services [43]. Additionally, social and cultural factors can influence ANC utilization. In upper-middle-income countries, there may be stronger social support systems and cultural norms that encourage women to seek ANC services. In contrast, women in low-income countries may face cultural barriers, such as traditional beliefs and practices that discourage the use of formal healthcare services. Accordingly, addressing the disparities in ANC utilization between low-income and upper-middle-income countries requires targeted interventions to improve healthcare infrastructure, financing, education, and cultural acceptance of ANC services.

### Limitation of the study

This study has several limitations that should be acknowledged. First, the reliance on self-reported data from the DHSs may introduce reporting biases, as participants might underreport or misreport their antenatal care utilization. Second, the cross-sectional design of the study limits the ability to establish causal relationships between identified factors and ANC non-utilization. Additionally, unmeasured confounding factors, such as the quality of healthcare services and women's perceptions of ANC, may also influence the results and warrant further investigation.

### Conclusion

A significant proportion of women in low- and middle-income countries do not utilize antenatal care services, which poses serious risks to maternal and child health. This study reveals that a combination of individual, community, and country-level factors contributes to this non-utilization. Barriers such as lack of education, socio-economic challenges, rural residence, and distance to healthcare facilities play crucial roles in limiting access to essential ANC services. To align with the WHO's recommendation of eight or more ANC contacts, policymakers should prioritize addressing these barriers.

 

Strategies could include enhancing education and awareness about the importance of ANC, improving infrastructure and transportation to health facilities, and increasing the availability of services in underserved areas.

## Supporting information

**S1 File. Minimal data.**
(ZIP)

## Author contributions

**Conceptualization:** Tadesse Tarik Tamir, Deresse Abebe Gebrehana, Alebachew Ferede Zegeye, Bewuketu Terefe, Berhan Tekeba.

**Data curation:** Tadesse Tarik Tamir, Deresse Abebe Gebrehana, Alebachew Ferede Zegeye, Bewuketu Terefe, Berhan Tekeba.

**Formal analysis:** Tadesse Tarik Tamir, Bewuketu Terefe.

**Investigation:** Tadesse Tarik Tamir, Deresse Abebe Gebrehana, Bewuketu Terefe, Berhan Tekeba.

**Methodology:** Tadesse Tarik Tamir, Alebachew Ferede Zegeye, Bewuketu Terefe, Berhan Tekeba.

**Software:** Tadesse Tarik Tamir.

**Supervision:** Tadesse Tarik Tamir.

**Validation:** Alebachew Ferede Zegeye, Bewuketu Terefe, Berhan Tekeba.

**Visualization:** Alebachew Ferede Zegeye, Bewuketu Terefe, Berhan Tekeba.

**Writing – original draft:** Tadesse Tarik Tamir.

**Writing – review & editing:** Tadesse Tarik Tamir, Deresse Abebe Gebrehana, Alebachew Ferede Zegeye, Bewuketu Terefe, Berhan Tekeba.

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
