## [Decision Letter · Decision Letter 0]

21 Jun 2025

Dear Dr. Tamir,

Thank you for submitting your manuscript to PLOS ONE. After careful consideration, we feel that it has merit but does not fully meet PLOS ONE’s publication criteria as it currently stands. Therefore, we invite you to submit a revised version of the manuscript that addresses the points raised during the review process.

Thank you for submitting this important and comprehensive paper to PLOS ONE journal. I appreciate the time and efforts authors have given in drafting this manuscript. The authors should revise their manuscript based on esteemed reviewers any my comments. 

We look forward to receiving your revised manuscript.

Kind regards,

Muhammad Haroon Stanikzai

Academic Editor

PLOS ONE

Journal Requirements:

3.We note that Figures 4 and 5 in your submission contain [map/satellite] images which may be copyrighted. All PLOS content is published under the Creative Commons Attribution License (CC BY 4.0), which means that the manuscript, images, and Supporting Information files will be freely available online, and any third party is permitted to access, download, copy, distribute, and use these materials in any way, even commercially, with proper attribution. For these reasons, we cannot publish previously copyrighted maps or satellite images created using proprietary data, such as Google software (Google Maps, Street View, and Earth). For more information, see our copyright guidelines: http://journals.plos.org/plosone/s/licenses-and-copyright.

a. You may seek permission from the original copyright holder of Figures 4 and 5 to publish the content specifically under the CC BY 4.0 license.

Additional Editor Comments :

- Please provide page and line number in the revision.

- Abstract: Please use adjusted odds ratio for significant factors in abstract as suggested by reviewer 1 too.

- Please use capital letters where needed. Only the first letter of headings and subheadings should be capital.

- Page 20-26 on PDF file: Is it low and middle countries or the author means low-and-middle income countries.

- I suggest the authors can use, cite, and consult recent publications published on ANC non-utilization in LMICs (https://journals.plos.org/plosone/article?id=10.1371/journal.pone.0309300, https://journals.plos.org/plosone/article?id=10.1371/journal.pone.0326130).

- As per journal requirements, the ethical approval should be mentioned in methods section.

- The references are not as per journal style as suggested by reviewer 2. Reference 5: Organization WH. I am not sure what does this mean. I believe the authors are using a software for reference managing. I ask them to manually edit them before submission as per journal style.

- The manuscript would benefit from language editing.

Reviewers' comments:

Reviewer's Responses to Questions

**Comments to the Author**

1. Is the manuscript technically sound, and do the data support the conclusions?

Reviewer #1: Yes

Reviewer #2: Yes

2. Has the statistical analysis been performed appropriately and rigorously?

Reviewer #1: Yes

Reviewer #2: I Don't Know

3. Have the authors made all data underlying the findings in their manuscript fully available?

Reviewer #1: Yes

Reviewer #2: Yes

4. Is the manuscript presented in an intelligible fashion and written in standard English?

Reviewer #1: Yes

Reviewer #2: Yes

Reviewer #1: Dear Editors and Authors,

Thank you for giving me the opportunity to review the manuscript, under the title “Magnitude, Geographic Distribution and Contributing Factors of Non-Utilization of Antenatal Care (ANC) Services among Women in Low- and Middle-Income Countries (LMICs): Insights for implementation of WHO recommendations”.

The authors chose a very important topic on maternal health care services in LMICs. This study examined data from DHSs (2015-2024) from 47 LMICs. The authors employed spatial analysis to show geographic distribution of ANC non-utilization and hierarchical analysis to identify factors associated with ANC non-utilization. The authors found that pooled magnitude of ANC non-utilization was at 10.59%, ranging 40.05% in Afghanistan to 0.76% in Burundi. Many regions in several countries identified as hotspots for ANC non-utilization. Factors significantly associated with higher odds of non-utilization were: no or low education, being unmarried, being in low wealth index, living in rural areas, distance to health facilities, and low-income levels of a country.

The manuscript is well written. The findings from this study have the potential to influence health interventions and policies to improve the use of ANC services in LMICs. The authors need to address the following issues before the manuscript can be considered for publication.

The full title seems a bit long. The authors may opt to use the short title and add few words to it.

In the abstract, the results should report the ORs (95%CI) for the factors significantly associated with non-utilization of ANC. If word count is an issue, the authors can summarize the prevalence of pooled non-utilization by stating for example, “Pooled prevalence of non-utilization of ANC among women in LMICs was at 10.59%, ranging from 40.05% in Afghanistan to 0.76% in Burundi.”, with many regions in several countries identified as hotspots for ANC non-utilization. Then, the authors can report the ORs for factors significantly associated with non-utilization of ANC.

Throughout the manuscript, the authors don’t need to write the full low- and middle-income countries. They just use it in full low- and middle-income countries (LMICs) when they use it first time in the manuscript; and then use the short term (LMICs) instead of the full term.

Under the sub-section “Magnitude of non-utilization of ANC services in LMICs”, the author should add in text that “Afghanistan had the highest non-utilization at 40.05% and Burundi had the lowest at 0.76%”.

For figure 4 (Hot spot analysis …) and figure 5 (Interpolation of non-utilization of ANC…), the authors should use a landscape of the maps as the current figures seem to be a bit confusing, showing the Asian countries (e.g., Tajikistan, Pakistan, India….) above Albania, Armenia, and Europe.

Reviewer #2: thank you for the opportunity to read your manuscript on a very important subject.

In LMICs, the issue is not only inadequate ANC visits, but also the timing of the ANC visits. do you have any information about timing of ANC to make your study stronger or not, if you have add this information otherwise just ignore it.

Most of the conditions studied are similar in LMICs, but why are some countries most affected? can you discuss more about it?

please, check all your references as there are some . fro example reference number 5.

**Do you want your identity to be public for this peer review?** For information about this choice, including consent withdrawal, please see our Privacy Policy

Reviewer #1: No

Reviewer #2: No

---

## [Author Response · Author response to Decision Letter 1]

16 Jul 2025

Response to comments

Subject: Submission of revised manuscript

Manuscript ID: (PONE-D-25-13560), [EMID:66dd1977a8a21886]

Title: Magnitude, Distribution and Determinants of Non-Utilization of Antenatal Care Services among Women in Low- and Middle-Income countries: Insights for Implementation of WHO recommendations

Journal: PLOS One

I hope this letter finds you well. We appreciate the diligent efforts of the editorial team in facilitating the review process for our manuscript. Additionally, we extend my gratitude to the editors and reviewers for their valuable time and thoughtful feedback, which significantly contributed to enhancing the quality of our work.

The constructive comments provided by the reviewers have been instrumental in refining our study. We are pleased to note that the reviewers share our assessment of the scientific significance of our findings. In response to their suggestions, we have meticulously addressed each point raised. Please find our comprehensive responses to the comments below.

Furthermore, I have attached the revised manuscript file separately for your convenience. We believe that the revisions strengthen the manuscript and align it more closely with the journal’s scope and standards.

Thank you for considering our work for publication. We hope that our revised submission meets the high standards set by PLOS One.

Best regards,

Corresponding Author

Response to Editor comments

- Please provide page and line number in the revision.

Response: Dear editor, thank you for your suggestion. We have provided page and line numbers in the revision.

- Abstract: Please use adjusted odds ratio for significant factors in abstract as suggested by reviewer 1 too.

Response: We have used adjusted odds ratio for significant factors in abstract as per the reviewer suggestion. Kindly find the point on page 2, lines 39 to 49 in our revised manuscript.

- Please use capital letters where needed. Only the first letter of headings and subheadings should be capital.

Response: Dear editor, thank you for your thoughtful comments and suggestions. We have made corrections to the headings and sub-headings as per your suggestion. Kindly find the point in our revised manuscript.

- Page 20-26 on PDF file: Is it low and middle countries or the author means low-and-middle income countries.

Response: Dear editor, thank you for the inquiry. We mean “low- and middle-income countries”

- I suggest the authors can use, cite, and consult recent publications published on ANC non-utilization in LMICs (https://journals.plos.org/plosone/article?id=10.1371/journal.pone.0309300, https://journals.plos.org/plosone/article?id=10.1371/journal.pone.0326130).

- As per journal requirements, the ethical approval should be mentioned in methods section.

Response: Dear Editor, we have mentioned the ethical approval should be mentioned in methods section as per the journal requirement based on your direction. Kindly see the page 10, lines 230-30 of our revised manuscript.

- The references are not as per journal style as suggested by reviewer 2. Reference 5: Organization WH. I am not sure what does this mean. I believe the authors are using a software for reference managing. I ask them to manually edit them before submission as per journal style.

Response: Dear editor, we have edited the references for alignment with journal style.

- The manuscript would benefit from language editing.

Response: Dear editor, thank you for your suggestion with regard to the language. We have edited the manuscript for language improvement. Thank you once again.

Response to reviewers comments

Reviewer #1: Dear Editors and Authors,

Thank you for giving me the opportunity to review the manuscript, under the title “Magnitude, Geographic Distribution and Contributing Factors of Non-Utilization of Antenatal Care (ANC) Services among Women in Low- and Middle-Income Countries (LMICs): Insights for implementation of WHO recommendations”.

The authors chose a very important topic on maternal health care services in LMICs. This study examined data from DHSs (2015-2024) from 47 LMICs. The authors employed spatial analysis to show geographic distribution of ANC non-utilization and hierarchical analysis to identify factors associated with ANC non-utilization. The authors found that pooled magnitude of ANC non-utilization was at 10.59%, ranging 40.05% in Afghanistan to 0.76% in Burundi. Many regions in several countries identified as hotspots for ANC non-utilization. Factors significantly associated with higher odds of non-utilization were: no or low education, being unmarried, being in low wealth index, living in rural areas, distance to health facilities, and low-income levels of a country.

The manuscript is well written. The findings from this study have the potential to influence health interventions and policies to improve the use of ANC services in LMICs. The authors need to address the following issues before the manuscript can be considered for publication.

Response: Dear reviewer, we sincerely appreciate your enthusiasm for our manuscript’s subject and hypotheses. Additionally, we value your detailed perspectives and insightful comments.

The full title seems a bit long. The authors may opt to use the short title and add few words to it.

Response: Dear reviewer, Thank you for your comments to our title formatting. We have made a kind of modification to reduce its length. Kindly find the modification in our revised manuscript.

In the abstract, the results should report the ORs (95%CI) for the factors significantly associated with non-utilization of ANC. If word count is an issue, the authors can summarize the prevalence of pooled non-utilization by stating for example, “Pooled prevalence of non-utilization of ANC among women in LMICs was at 10.59%, ranging from 40.05% in Afghanistan to 0.76% in Burundi.”, with many regions in several countries identified as hotspots for ANC non-utilization. Then, the authors can report the ORs for factors significantly associated with non-utilization of ANC.

Response: Dear reviewer, thank you for your insightful correction and guidance on our result writeup. We have added the AORs with its 95 CI in our revised manuscript as per our suggestion. Kindly find the point on page 2, lines 39 to 49 in our revised manuscript.

Throughout the manuscript, the authors don’t need to write the full low- and middle-income countries. They just use it in full low- and middle-income countries (LMICs) when they use it first time in the manuscript; and then use the short term (LMICs) instead of the full term.

Response: Dear reviewer, thank you for your detailed review of our work and valuable corrections. We have followed your direction in writing our revised manuscript.

Under the sub-section “Magnitude of non-utilization of ANC services in LMICs”, the author should add in text that “Afghanistan had the highest non-utilization at 40.05% and Burundi had the lowest at 0.76%”.

Response: Dear reviewer, thank you for your correction and direction on our findings. We have added the points you suggested in our revised manuscript. Kindly find the point on page 13, line 277-278 of our revised manuscript.

For figure 4 (Hot spot analysis …) and figure 5 (Interpolation of non-utilization of ANC…), the authors should use a landscape of the maps as the current figures seem to be a bit confusing, showing the Asian countries (e.g., Tajikistan, Pakistan, India….) above Albania, Armenia, and Europe.

Response: Dear reviewer, you are correct. We have made the replacements as per your suggestion. Kindly see our new figures on page 17 and 18 of our revised manuscript for the amendments.

Reviewer #2: thank you for the opportunity to read your manuscript on a very important subject.

In LMICs, the issue is not only inadequate ANC visits, but also the timing of the ANC visits. do you have any information about timing of ANC to make your study stronger or not, if you have add this information otherwise just ignore it.

Response: Dear reviewer, thank you for your insightful suggestions on our manuscript. You are correct, it is not only the inadequate ANC visits which is concerning, but also the timing. However, we observed that the timing of ANC visit was not commonly reported in those countries and that was main reason for its exclusion in this study.

Most of the conditions studied are similar in LMICs, but why are some countries most affected? can you discuss more about it?

Response: Thank you for your invaluable feedback. We have discussed the marked variation in the service utilization accross countries. Kindly find the point in our discussion, lines 445-464 in revised manuscript.

please, check all your references as there are some. for example, reference number 5.

Response: Dear reviewer, thank you for the thoughtful feedback. We have checked all of the references in our revised manuscript as per your suggestion. Kindly check our references in the revised manuscript.

Thank you once again.

---

## [Editor Report · Decision Letter 1]

5 Aug 2025

Magnitude, Distribution and Determinants of Non-Utilization of Antenatal Care Services among Women in Low- and Middle-Income countries: Insights for Implementation of WHO Recommendations

PONE-D-25-13560R1

Dear Dr. Tamir,

We’re pleased to inform you that your manuscript has been judged scientifically suitable for publication and will be formally accepted for publication once it meets all outstanding technical requirements.

Kind regards,

Muhammad Haroon Stanikzai

Academic Editor

PLOS ONE

Journal office comments: We note that the academic editor has recommended that you cite specific previously published works. As always, we recommend that you please review and evaluate the requested works to determine whether they are relevant and should be cited. It is not a requirement to cite these works.

Additional Editor Comments (optional):

Thank you for your diligent edits to the manuscript.

Please carefully proofread the entire manuscript at Proof correction.

- Abstract: Line 54: low-and middle-income countries.

- Line 542: Please remove 1 after services.

- Line 658: Please make the first letter of study in lower-case.
---

## [Editor Report · Acceptance letter]

PONE-D-25-13560R1

PLOS ONE

Dear Dr. Tamir,

I'm pleased to inform you that your manuscript has been deemed suitable for publication in PLOS ONE. Congratulations! Your manuscript is now being handed over to our production team.

Kind regards,

on behalf of

Dr. Muhammad Haroon Stanikzai

Academic Editor

PLOS ONE